# *In Vitro* Susceptibilities of Worldwide Isolates of Intrapulmonary *Aspergillus* Species and Important *Candida* Species in Sterile Body Sites against Important Antifungals: Data from the Antimicrobial Testing Leadership and Surveillance Program, 2017–2020

Shio-Shin Jean,[a,b,i] Hung-Jen Yang,[c] Po-Chuen Hsieh,[b] Yu-Tsung Huang,[d,j] Wen-Chien Ko,[e] Po-Ren Hsueh[d,f,g,h,j,k]

[a]Department of Internal Medicine, Min-Sheng General Hospital, Taoyuan, Taiwan

[b]Department of Pharmacy, College of Pharmacy and Health Care, Tajen University, Pingtung, Taiwan

[c]Department of Family Medicine, Min-Sheng General Hospital, Taoyuan, Taiwan

[d]Department of Laboratory Medicine, National Taiwan University Hospital, National Taiwan University College of Medicine, Taipei, Taiwan

[e]Department of Medicine, College of Medicine, National Cheng Kung University, Tainan, Taiwan

[f]Department of Laboratory Medicine, China Medical University Hospital, China Medical University, Taichung, Taiwan

[g]School of Medicine, China Medical University, Taichung, Taiwan

[h]Ph.D Program for Aging, School of Medicine, China Medical University, Taichung, Taiwan

[i]Department of Critical Care Medicine, Min-Sheng General Hospital, Taoyuan, Taiwan

[j]Department of Internal Medicine, National Taiwan University Hospital, National Taiwan University College of Medicine, Taipei, Taiwan

[k]Department of Internal Medicine, China Medical University Hospital, China Medical University, Taichung, Taiwan

**ABSTRACT** To understand the changes of resistance in clinically commonly encountered fungi, we used the Antimicrobial Testing Leadership and Surveillance (ATLAS) database to explore *in vitro* antifungal susceptibilities against clinically important isolates of *Aspergillus* and *Candida* species (collected from intrapulmonary and sterile body areas, respectively). We applied the CLSI antifungal 2020 and the EUCAST antifungal 2020 guidelines. From 2017 to 2020, isolates of intrapulmonary *Aspergillus fumigatus* ($n = 660$), *Aspergillus niger* ($n = 107$), *Aspergillus flavus* ($n = 96$), *Aspergillus terreus* ($n = 40$), and *Aspergillus nidulans* species complex ($n = 26$) and sterile site-originated isolates of *Candida albicans* ($n = 1,810$), *Candida glabrata* ($n = 894$), *Candida krusei* ($n = 120$), *Candida dubliniensis* ($n = 107$), *Candida lusitaniae* ($n = 82$), *Candida guilliermondii* ($n = 28$), and *Candida auris* ($n = 7$) were enrolled in this study. Using the EUCAST 2020 breakpoints, it was demonstrated that amphotericin B and posaconazole displayed poor *in vitro* susceptibility rates against *A. fumigatus* isolates ($<50\%$ and 18.9%, respectively). In contrast, isavuconazole and itraconazole showed high *in vitro* potency against most *Aspergillus* isolates ($>92\%$). Most intrapulmonary *Aspergillus* isolates exhibited MICs of $\leq 0.06$ $\mu$g/mL to anidulafungin. Furthermore, intrapulmonary *A. fumigatus* isolates collected from Italy and the United Kingdom exhibited lower *in vitro* susceptibility to isavuconazole (72.2% and 69%, respectively) than those in the remaining ATLAS participant countries ($>85\%$). Higher isavuconazole $MIC_{90}$s against *C. auris* and *C. guilliermondii* (1 and 4 $\mu$g/mL, respectively) were observed compared to the other five *Candida* species. Despite the aforementioned MICs and susceptibilities against fungi, research needs to consider the pharmacokinetic (PK) profiles, pharmacodynamic (PD) parameters, and clinical treatment experience with antifungals against specific *Aspergillus* species.

**IMPORTANCE** In addition to monitoring the antifungal susceptibilities of clinically important fungi, reviewing the PK/PD indices and the clinical therapy experience of antifungals under evaluation are important to guide an appropriate antifungal prescription. The efficacies of liposomal amphotericin B complex and anidulafungin for the treatment of

Address correspondence to Po-Ren Hsueh, hsporen@gmail.com.

The authors declare no conflict of interest.

pulmonary aspergillosis caused by different *Aspergillus* species need to be periodically evaluated in the future.

**KEYWORDS** intrapulmonary *Aspergillus* species, *Candida* species, sterile body sites, isavuconazole, amphotericin B, anidulafungin, ATLAS

Fungal infections can cause life-threatening complications. Among the various kinds of fungal infections, invasive candidiasis and candidemia were ranked among the top 10 nosocomial infections in the last decade (1) and are notable invasive fungal diseases (IFD) that frequently cause high mortality rates ranging from 30% to 50% (2, 3). Similarly, invasive pulmonary aspergillosis (IPA) has been reported to result in morbidities; particularly, 27.4% to 90% of IPA cases result in fatality, depending on the IPA severity and immune condition of patients (4–6). The incidence of IPA is increasing with the growing number of immunocompromised hosts, prescription of broad-spectrum antibacterial antibiotic(s) for prolonged durations in intensive care units, and patients with severe viral infections, such as influenza and COVID-19 (6–8). A multicenter study (predominantly in many member states of the European Union [EU]) revealed that triazole-resistant *Aspergillus fumigatus* isolates collected from 2009 to 2011 were more frequent (prevalence of 3.2%) than previously acknowledged (9). Periodic monitoring of fungal species-specific susceptibility is beneficial for understanding changes in antifungal resistance and recommending appropriate antifungal agents against clinically important fungi. However, substantially fewer surveys on antifungal resistance have been performed than those aimed at the resistance of bacteria of interest.

Antifungal armamentaria against clinically important fungi have been developing over the past 2 decades. According to the latest European Society for Clinical Microbiology and Infectious Diseases (ESCMID), the European Confederation of Medical Mycology (ECMM), and the European Respiratory Society (ERS) joint clinical guidelines, voriconazole (VRC) and isavuconazole (ISA) are currently the first-line treatment against IPA. In addition, posaconazole (POS) is recommended for antifungal prophylaxis during prolonged neutropenia in high-risk populations or as salvage therapy for nonresponsive patients. In contrast, liposomal amphotericin B (AMB) is recommended as salvage therapy against antimold triazole-resistant *Aspergillus* isolates (10). Using the antifungal database in the Antimicrobial Testing Leadership and Surveillance (ATLAS), guidelines of the Clinical and Laboratory Standards Institute (CLSI) recommended in 2020 (regarding *Candida* species and VRC against *A. fumigatus*) (11), and European Committee on Antimicrobial Susceptibility Testing (EUCAST) 2020 guidelines (12), we investigated the distributions of MICs of important antifungals, and the evolutionary trend of antifungal susceptibility results against clinically important fungi worldwide (intrapulmonary IPA and some important *Candida* species [13] cultured from sterile body sites) collected from 2017 through 2020. Moreover, pharmacokinetic (PK) and pharmacodynamic (PD) investigations of each antibiotic are crucial for defining the optimal antimicrobial exposure to determine its *in vitro* potency against the organisms under evaluation (7, 14). Therefore, we also assessed the clinical feasibility of POS, anidulafungin (AFG), and other antifungals for treating IPA isolates using the MIC distributions in this survey, PK profiles, and PD parameters of the antifungal agents.

## RESULTS

**Number of isolates of *Aspergillus* and *Candida* species.** In the 2017 to 2020 ATLAS antifungal program, 929 intrapulmonary isolates of various important *Aspergillus* species were collected from the following four regions: Europe (many EU member states, the United Kingdom, and Turkey; 443 isolates), North America (the United States and Canada; 328 isolates), Asia-Pacific (five countries; 143 isolates), and Latin America (Brazil only; 15 isolates). Furthermore, susceptibility data of 3,048 isolates of potentially high-resistance *Candida* species were collected from the following four regions: Europe (some EU member states, Switzerland, Turkey, Israel, and the United Kingdom; 1,432 isolates), North America (two countries; 951 isolates), Asia-Pacific (six countries; 401 isolates), and Latin America (five countries; 264 isolates).

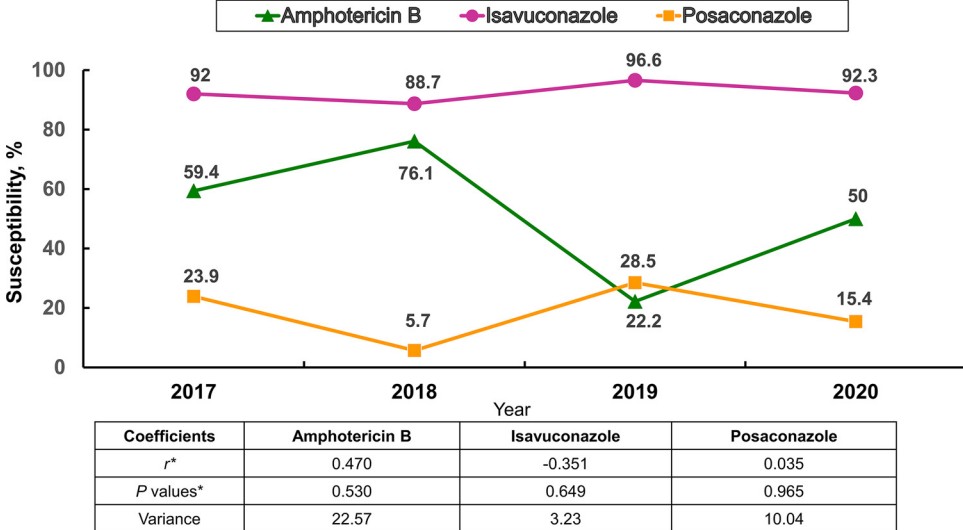

| Coefficients | Amphotericin B | Isavuconazole | Posaconazole |
|---|---|---|---|
| *r** | 0.470 | -0.351 | 0.035 |
| *P* values* | 0.530 | 0.649 | 0.965 |
| Variance | 22.57 | 3.23 | 10.04 |

*Analyzed by Pearson's correlation

**FIG 1** Trend of annual *in vitro* susceptibility rates of amphotericin B, isavuconazole, and posaconazole against 660 intrapulmonary *Aspergillus fumigatus* isolates collected from 2017 through 2020 in the ATLAS antifungal database using the EUCAST 2020 guidelines.

**Susceptibility data of isolates of intrapulmonary *Aspergillus* spp. against antifungals.** Figure 1 illustrates the changes in the annual *in vitro* susceptibility rates of AMB, ISA, and POS against 660 intrapulmonary *A. fumigatus* isolates collected from 2017 to 2020 in the ATLAS antifungal database using the EUCAST 2020 guidelines. During the 4-year study period, as assessed by the EUCAST 2020 guidelines, a wider fluctuation in the annual susceptibility rates of AMB against isolates of *A. fumigatus* than those of ISA and POS was observed in terms of gross variance.

Table 1 illustrates the distributions of MIC and susceptibility data for some antifungal agents (AMB and four antimold triazoles) against worldwide intrapulmonary isolates of various important *Aspergillus* species (including *A. fumigatus* [*n* = 660], *Aspergillus flavus* species complex [*n* = 96], *Aspergillus nidulans* species complex [*n* = 26], *Aspergillus niger* species complex [*n* = 107], and *Aspergillus terreus* species complex [*n* = 40]) in the 2017 to 2020 ATLAS antifungal database. The susceptibility rates and $MIC_{50/90}$ values of AMB against 660 *A. fumigatus* isolates were 49.5% as evaluated by the EUCAST 2020 guidelines and 2 and 2 $\mu$g/mL, respectively. Among 22 (3.3%) VRC-resistant *A. fumigatus* isolates (MICs of VRC, $\geq$2 $\mu$g/mL), 12 (54.5%) and 19 (86.4%) isolates exhibited notable resistance *in vitro* to AMB and ISA, respectively. In striking contrast with *A. fumigatus* isolates, the *in vitro* susceptibility of AMB against isolates of *A. niger* species complex (96.3%, assessed by the EUCAST 2020 guidelines) was higher than that against *A. fumigatus* ($P < 0.001$; $MIC_{50/90}$ of AMB, 0.5 and 1 $\mu$g/mL, respectively). The $MIC_{50/90}$ values of AMB against isolates of *A. flavus*, *A. nidulans*, and *A. terreus* species complex enrolled in this survey were 2 and 2, 2 and 2, and 2 and 4 $\mu$g/mL, respectively.

ISA was shown to exhibit >92% rates of *in vitro* susceptibility against isolates of *A. fumigatus*, *A. flavus*, *A. nidulans*, and *A. terreus* species complex according to the EUCAST 2020 guidelines. Similar to ISA, isolates of most *Aspergillus* species exhibited 93 to 100% susceptibilities to itraconazole (ITC) using the EUCAST 2020 guidelines, except for *A. niger* species complex, which was due to the lack of clinical breakpoints. Furthermore, the susceptibility rates of VRC against isolates of *A. fumigatus* were 92.3% and 96.7%, respectively, when assessed using the CLSI 2020 and EUCAST 2020 guidelines. An excellent *in vitro* susceptibility to VRC was also observed in isolates of *A. nidulans* species complex when applying the EUCAST 2020 guidelines. Despite the lack of susceptibility breakpoints, we observed that the distributions of MIC values of VRC against *A. terreus* and *A. flavus* species complex were significantly lower than those of

**TABLE 1** Distributions of MICs and susceptibility profiles of various antifungal agents (polyene, four antimold triazoles) against intrapulmonary isolates of *Aspergillus fumigatus*, *Aspergillus nidulans*, *Aspergillus terreus*, *Aspergillus niger*, and *Aspergillus flavus* species complex cultured from 2017 through 2020 in the Antimicrobial Testing Leadership and Surveillance project according to the guidelines of the Clinical and Laboratory Standards Institute 2020 (12) and the European Committee on Antimicrobial Susceptibility Testing 2020 (13)[c]

| Species (no. of isolates) and antifungal agent | No. of isolates (cumulative percentage) according to MIC (µg/mL) | | | | | | | | | | MIC$_{50}$ (µg/mL) | MIC$_{90}$ (µg/mL) | S (%) | I (%) | R (%) |
|---|---|---|---|---|---|---|---|---|---|---|---|---|---|---|---|
| | 0.03 | 0.06 | 0.12 | 0.25 | 0.5 | 1 | 2 | 4 | 8 | 16 | | | | | |
| **A. fumigatus (660)** | | | | | | | | | | | | | | | |
| Amphotericin B | | | | 1 (0.2) | 11 (1.8) | 315 (49.5) | **331 (99.7)** | 2 (100) | | | 2 | 2 | 49.5[b] | N/A | 50.5[b] |
| Amphotericin B, if MICs of voriconazole >2 µg/mL (22) | | | | | | 10 (45.5) | **12 (100)** | | | | 2 | 2 | 45.5[b] | N/A | 54.5[b] |
| Isavuconazole | | | 3 (0.5) | 49 (7.9) | 422 (71.8) | **138 (92.7)** | 26 (96.7) | 13 (98.6) | 5 (99.4) | 4 (100) | 0.5 | 1 | 92.7[b] | N/A | 3.3[b] |
| Isavuconazole, if MICs of voriconazole >2 µg/mL (22) | | | | | | 1 (4.5) | 2 (13.6) | 10 (59.1) | 5 (81.8) | **4 (100)** | 4 | 16 | 4.5[b] | N/A | 86.4[b] |
| Voriconazole | | 1 (0.2) | 13 (2.1) | 311 (49.2) | **284 (92.3)** | 29 (96.7) | 16 (99.1) | 3 (99.5) | 1 (99.7) | 2 (100) | 0.5 | 0.5 | 92.3[a]/96.7[b] | 4.4[a] | 3.3[a,b] |
| Posaconazole | | 7 (1.1) | 118 (18.9) | 331 (69.1) | **186 (97.3)** | 16 (99.7) | 0 (99.7) | | 1 (100) | | 0.25 | 0.5 | 18.9[b] | N/A | 30.9[b] |
| Itraconazole | | | | 36 (5.5) | 295 (50.2) | **282 (92.9)** | 27 (97) | 10 (98.5) | 4 (99.1) | 6 (100) | 0.5 | 1 | 92.9[b] | N/A | 7.1[b] |
| *Aspergillus nidulans* spp. complex (26) | | | | | | | | | | | | | | | |
| Amphotericin B | | | | | 2 (7.7) | 5 (26.9) | **18 (96.2)** | 0 (96.2) | 1 (100) | | 2 | 2 | N/A | N/A | N/A |
| Isavuconazole | 3 (11.5) | 4 (26.9) | 11 (69.2) | **6 (92.3)** | 2 (100) | | | | | | 0.25 | 0.25 | 100[b] | N/A | 0[b] |
| Voriconazole | 2 (7.7) | 4 (23.1) | 13 (73.1) | **7 (100)** | | | | | | | 0.12 | 0.25 | 100[b] | N/A | 0[b] |
| Posaconazole | | 1 (3.8) | 6 (26.9) | 11 (69.2) | **8 (100)** | | | | | | 0.25 | 0.5 | N/A | N/A | N/A |
| Itraconazole | | | 2 (7.7) | 4 (23.1) | 15 (80.8) | **5 (100)** | | | | | 0.5 | 1 | 100[b] | N/A | 0[b] |
| *Aspergillus terreus* spp. complex (40) | | | | | | | | | | | | | | | |
| Amphotericin B | | | | | | 9 (22.5) | 25 (85) | **6 (100)** | | | 2 | 4 | N/A | N/A | N/A |
| Isavuconazole | | 1 (2.5) | 7 (20) | 10 (45) | **20 (95)** | 1 (97.5) | 1 (100) | | | | 0.5 | 0.5 | 97.5[b] | N/A | 0[b] |
| Voriconazole | | 1 (2.5) | 7 (20) | 12 (50) | **19 (97.5)** | 1 (100) | | | | | 0.25 | 0.5 | N/A | N/A | N/A |
| Posaconazole | | | 5 (12.5) | 27 (80) | **8 (100)** | | | | | | 0.25 | 0.5 | 12.5[b] | N/A | 20[b] |
| Itraconazole | | | 1 (2.5) | 8 (22.5) | 26 (87.5) | **5 (100)** | | | | | 0.5 | 1 | 100[b] | N/A | 0[b] |
| *Aspergillus niger* spp. complex (107) | | | | | | | | | | | | | | | |
| Amphotericin B | | | 1 (0.9) | 4 (4.7) | 62 (62.6) | **36 (96.3)** | 4 (100) | | | | 0.5 | 1 | 96.3[b] | N/A | 2.7[b] |
| Isavuconazole | | | | | 9 (8.4) | 45 (50.5) | 29 (77.6) | **19 (95.3)** | 3 (98.1) | 2 (100) | 1 | 4 | N/A | N/A | N/A |
| Voriconazole | | | 2 (1.9) | 3 (4.7) | 47 (48.6) | 39 (85) | **15 (99.1)** | 1 (100) | | | 1 | 2 | N/A | N/A | N/A |
| Posaconazole | | | 2 (1.9) | 9 (10.3) | 76 (81.3) | **20 (100)** | | | | | 0.5 | 1 | N/A | N/A | N/A |
| Itraconazole | | | 1 (0.9) | 2 (2.8) | 5 (7.5) | 34 (39.3) | 51 (86.9) | **10 (96.3)** | 4 (100) | | 2 | 4 | N/A | N/A | N/A |
| *Aspergillus flavus* spp. complex (96) | | | | | | | | | | | | | | | |
| Amphotericin B | | | | | | 23 (24) | **67 (93.8)** | 5 (99) | 1 (100) | | 2 | 2 | N/A | N/A | N/A |
| Isavuconazole | | | 1 (1) | 13 (14.6) | 50 (66.7) | **31 (99)** | 1 (100) | | | | 0.5 | 1 | 99[b] | N/A | 0[b] |
| Voriconazole | | | | 22 (22.9) | 52 (77.1) | **21 (99)** | 1 (100) | | | | 0.5 | 1 | N/A | N/A | N/A |
| Posaconazole | | | 7 (7.3) | 39 (47.9) | **49 (99)** | 1 (100) | | | | | 0.5 | 0.5 | N/A | N/A | N/A |
| Itraconazole | | | 1 (1) | 12 (13.5) | 54 (69.8) | **29 (100)** | | | | | 0.5 | 1 | 100[b] | N/A | 0[b] |

[a]According to the clinical breakpoints recommended by the Clinical and Laboratory Standards Institute 2020.
[b]According to the clinical breakpoints recommended by the European Committee on Antimicrobial Susceptibility Testing 2020.
[c]Bold values in parentheses indicate the numerals of cumulative percentage above 90% of isolates for given subsets. S, susceptible; I, intermediate; R, resistant; N/A, not applicable.

**TABLE 2** Distributions of MICs and $MIC_{50}$ and $MIC_{90}$ values of anidulafungin against worldwide intrapulmonary isolates of *Aspergillus fumigatus*, *Aspergillus nidulans*, *Aspergillus terreus*, *Aspergillus niger*, and *Aspergillus flavus* species complex cultured from 2017 through 2020 in the Antimicrobial Testing Leadership and Surveillance project[a]

| *Aspergillus* spp. (no. of isolates) | No. of isolates (cumulative %) according to MIC ($\mu$g/mL) | | | | | | | $MIC_{50}/MIC_{90}$ ($\mu$g/mL) |
|---|---|---|---|---|---|---|---|---|
| | 0.002 | 0.004 | 0.008 | 0.016 | 0.03 | 0.06 | 0.12 | |
| *Aspergillus fumigatus* (660) | 26 (3.9) | 91 (17.7) | 167 (43) | 241 (79.5) | 110 (**96.2**) | 25 (100) | | 0016/0.03 |
| *Aspergillus fumigatus* if MICs of VRC >2 $\mu$g/mL (22) | 2 (9.1) | 3 (22.7) | 8 (59.1) | 6 (86.4) | 2 (**95.5**) | 1 (100) | | 0.008/0.03 |
| *Aspergillus nidulans* spp. complex (26) | 2 (7.7) | 4 (23.1) | 5 (42.3) | 11 (84.6) | 3 (**96.2**) | 0 (96.2) | 1 (100) | 0.016/0.03 |
| *Aspergillus terreus* spp. complex (40) | 3 (7.5) | 4 (17.5) | 10 (42.5) | 15 (80) | 7 (**97.5**) | 1 (100) | | 0.016/0.03 |
| *Aspergillus niger* spp. complex (107) | 21 (19.6) | 40 (57) | 32 (86.9) | 11 (**97.2**) | 3 (100) | | | 0.004/0.016 |
| *Aspergillus flavus* spp. complex (96) | 12 (12.5) | 34 (47.9) | 30 (79.2) | 18 (**97.9**) | 2 (100) | | | 0.008/0.016 |

[a]Bold values in parentheses indicate the numerals of cumulative percentage above 90% of isolates for given subsets.

*A. niger* species complex ($P < 0.001$, between different *Aspergillus* species as assessed using the Mann-Whitney U test). The $MIC_{50/90}$ values of VRC against *A. terreus*, *A. flavus*, and *A. niger* species complex were 0.25 and 0.5, 0.5 and 1, and 1 and 2 $\mu$g/mL, respectively. In addition, POS was demonstrated to be active *in vitro* against only 18.9% and 12.5% of intrapulmonary isolates of *A. fumigatus* and *A. terreus* species complex, respectively, when applying the EUCAST 2020 guidelines.

**The MIC distributions of AFG against isolates of various *Aspergillus* species.** Table 2 illustrates the MIC distributions and $MIC_{50/90}$ values of AFG against worldwide intrapulmonary isolates of different *Aspergillus* species in the 2017 to 2020 ATLAS antifungal database. The $MIC_{90}$s of AFG against all *Aspergillus* species ranged from 0.016 to 0.03 $\mu$g/mL. In addition, regardless of VRC resistance, similar MIC distributions of AFG against *A. fumigatus* isolates were observed.

**Susceptibility rates of AMB and ISA against *A. fumigatus* collected from different ATLAS participant countries.** Figure 2 shows the susceptibility rates of AMB and ISA against isolates of *A. fumigatus* among 15 ATLAS participant countries that submitted at least 10 isolates from 2017 to 2020 as evaluated using the EUCAST 2020 guidelines. In brief, except in France (76.2%), Czechia (71.4%), Turkey (68%), Australia (64.8%), and Thailand (60%), the susceptibility rates of AMB against intrapulmonary isolates of *A. fumigatus* collected from the remaining 10 countries were less than 60% when evaluated

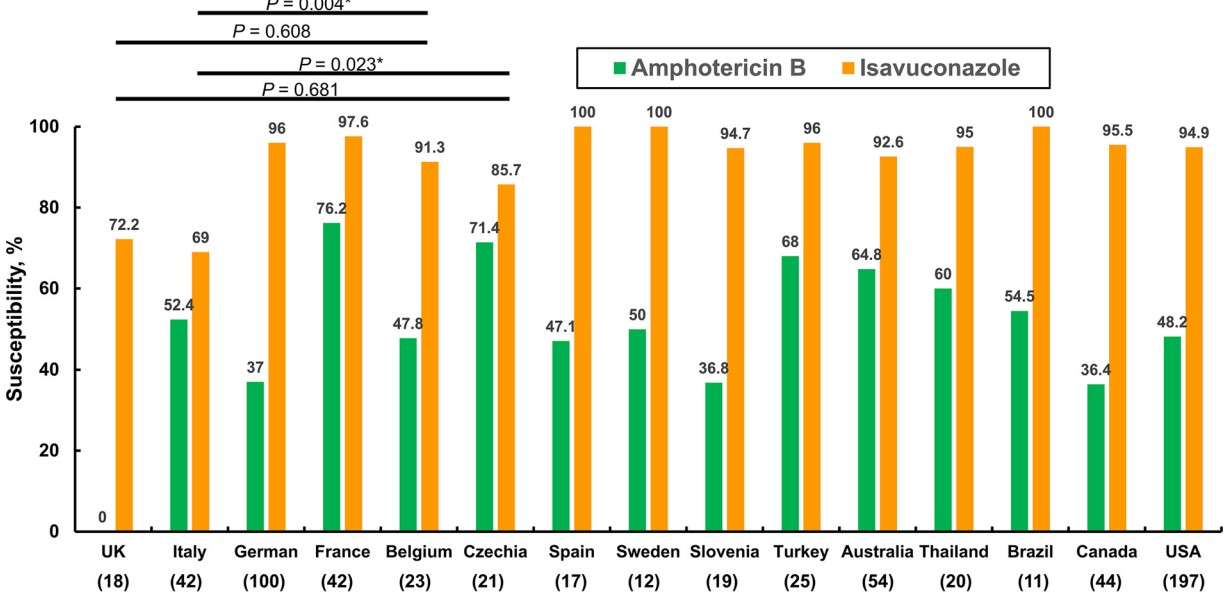

**FIG 2** *In vitro* susceptibility rates of amphotericin B and isavuconazole against intrapulmonary *Aspergillus fumigatus* isolates collected from 2017 through 2020 in the ATLAS antifungal database using the EUCAST 2020 guidelines. Differences in the susceptibility rates of isavuconazole against *A. fumigatus* isolates cultured in some countries were analyzed using the Mann-Whitney U test. *, $P$ value of <0.05 (the numeral in parenthesis indicates the number of isolates).

using the EUCAST 2020 guidelines. In contrast, with the exception of susceptibility to ISA in Italy (69%), the United Kingdom (72.2%), and Czechia (85.7%), those against *A. fumigatus* isolates collected from the remaining 12 participant countries were higher than 91% when evaluated using the EUCAST 2020 guidelines. The ISA MIC distribution of intrapulmonary *A. fumigatus* isolates collected in Italy was significantly higher than those of other countries ($P < 0.05$), except for the United Kingdom (Fig. 2), when evaluated using the Mann-Whitney U test.

**Susceptibility rates of antifungals against important *Candida* spp.** Table 3 presents the MIC distributions of four antimold triazoles, fluconazole (FLC), AFG, and AMB against 3,048 isolates worldwide of various important *Candida* species—including *Candida albicans* ($n = 1,810$ [59.4%]), *Candida glabrata* ($n = 894$ [29.3%]), *Candida krusei* ($n = 120$ [3.9%]), *Candida dubliniensis* ($n = 107$ [3.5%]), *Candida lusitaniae* ($n = 82$ [2.7%]), *Candida guilliermondii* ($n = 28$ [0.9%]), and *Candida auris* ($n = 7$ [0.23%])—cultured from sterile body sites in the 2017 to 2020 ATLAS database. Clinical breakpoints for interpreting the susceptibilities of some antifungal agents to *Candida* species are currently lacking. Using the EUCAST 2020 guidelines, the susceptibility rates of POS against isolates of *C. albicans* and *C. dubliniensis* were 97.6% and 97.2%, respectively. The susceptibility rates of VRC against isolates of *C. albicans* and *C. krusei* were 99.9% and 99.2%, respectively, when applying the CLSI 2020 guidelines. In addition, the MIC$_{90}$ values of ISA were equal to or 2-fold higher than those of VRC against the isolates of various *Candida* species analyzed in this study. Among different *Candida* species, the susceptibility rates of AMB against isolates of *C. albicans*, *C. dubliniensis*, *C. glabrata*, and *C. krusei* were 99.9%, 100%, 99.8%, and 85.8%, respectively, when applying the EUCAST 2020 criteria. Moreover, between the guidelines of CLSI 2020 and EUCAST 2020, we observed a more significant discrepancy in the susceptibility rates ($P < 0.001$) of AFG against isolates of *C. glabrata* (27.4% [95.6% versus 68.2%]) than those of *C. albicans* (12.1% [99.9% versus 87.8%]) and *C. krusei* (13.3% [100% versus 86.7%]). Figure 3 illustrates the *in vitro* susceptibility rates of four antifungals against isolates of important *Candida* species in the 2017 to 2020 ATLAS antifungal database using the EUCAST 2020 guidelines.

## DISCUSSION

This 2017 to 2020 antifungal ATLAS study highlighted several important points. First, using the EUCAST 2020 breakpoints, AMB and POS were significantly less active *in vitro* against intrapulmonary *A. fumigatus* than other antimold triazoles. In contrast, ISA and ITC were highly active *in vitro* against the isolates of many *Aspergillus* species. Second, most intrapulmonary *Aspergillus* isolates exhibited AFG MICs of $\leq 0.06$ $\mu$g/mL. Third, the intrapulmonary *A. fumigatus* isolates collected from Italy and the United Kingdom exhibited higher *in vitro* nonsusceptibility (NS) rates to ISA than those in the remaining ATLAS participant countries, which has never been addressed in the literature. Fourth, higher ISA MIC$_{90}$s were observed against isolates of *C. auris* and *C. guilliermondii* (1 and 4 $\mu$g/mL, respectively) than against the other five *Candida* species (ranging from 0.008 to 0.5 $\mu$g/mL). Table 4 summarizes the susceptible breakpoints of different antifungals against various *Aspergillus* and *Candida* species in the CLSI 2020 and EUCAST 2020 guidelines (12, 13). Additionally, Table 5 summarizes the PK/PD profiles of different antimold antifungal agents against molds. We utilized these data and mathematical calculations to estimate the suitability of respective antifungal agent against these important fungi.

In agreement with a 2007 to 2008 Korean study investigating the *in vitro* antifungal susceptibilities of 636 bloodstream isolates of *Candida* species (including *C. albicans* and *C. glabrata*) (15), the present susceptibility data showed that isolates of four *Candida* species exhibited excellent susceptibility rates to AMB and VRC ($>85\%$ and $\geq 99\%$, respectively) using the current guidelines. In addition, the FLC MIC$_{90}$s against isolates of *C. guilliermondii* and *C. lusitaniae* (16 and 1 $\mu$g/mL, respectively) were the same as those of the 2011 to 2016 Taiwanese survey (16).

A 2013 Korean study of 56 *A. fumigatus* isolates showed that the *in vitro* susceptibility rate to AMB was 91.1% (17), which was similar to that of a 2018 to 2021 Iranian

**TABLE 3** MIC distributions of various antimold triazoles, anidulafungin, and amphotericin B against worldwide isolates of different *Candida* species collected from sterile body sites from 2017 through 2020 in the Antimicrobial Testing Leadership and Surveillance project

| Candida spp. and antifungal agent | No. of isolates (cumulative %) according to MIC (µg/mL) | | | | | | | | | | | | | | MIC₉₀ (µg/mL) |
|---|---|---|---|---|---|---|---|---|---|---|---|---|---|---|---|
| | 0.002 | 0.004 | 0.008 | 0.015 | 0.03 | 0.06 | 0.12 | 0.25 | 0.5 | 1 | 2 | 4 | 8 | >16 | |
| **Candida albicans (1,810)** | | | | | | | | | | | | | | | |
| VRC | 193 (10.7) | 645 (46.3) | 749 (87.7) | 167 (96.9) | 46 (99.4) | 8 (99.9) | 0 (99.9) | 0 (99.9) | 1 (99.9) | 0 (99.9) | 0 (99.9) | 0 (99.9) | 1 (100) | | 0.015 |
| POS | 2 (0.1) | 20 (1.2) | 55 (4.3) | 365 (24.4) | 882 (73.1) | 443 (97.6) | 40 (99.8) | 1 (99.9) | 1 (99.9) | 0 (99.9) | 0 (99.9) | 0 (99.9) | 1 (100) | | 0.06 |
| ISA | 574 (31.7) | 544 (61.8) | 577 (93.6) | 93 (98.8) | 15 (99.6) | 4 (99.8) | 0 (99.8) | 2 (99.9) | 0 (99.9) | 0 (99.9) | 0 (99.9) | 0 (99.9) | 1 (100) | | 0.008 |
| FLC | | | 2 (0.1) | 11 (0.7) | 11 (1.3) | 160 (10.2) | 1168 (74.7) | 363 (94.8) | 67 (98.5) | 19 (99.5) | 5 (99.8) | 1 (99.8) | 1 (99.9) | 2 (100) | 0.25 |
| ITC | 1 (0.06) | 25 (1.4) | 35 (3.4) | 135 (10.8) | 371 (31.3) | 840 (77.7) | 349 (97) | 49 (99.7) | 3 (99.9) | 1 (99.9) | 0 (99.9) | 0 (99.9) | 1 (100) | | 0.12 |
| AFG | 37 (2) | 148 (10.2) | 433 (34.1) | 523 (63) | 449 (87.8) | 195 (98.6) | 22 (99.8) | 2 (99.9) | 0 (99.9) | 1 (100) | | | | | 0.06 |
| AMB | | | | | | 1 (0.06) | 1 (0.1) | 125 (7) | 1435 (86.3) | 246 (99.9) | 2 (100) | | | | 1 |
| **Candida glabrata (894)** | | | | | | | | | | | | | | | |
| VRC | | | 2 (0.2) | 9 (1.2) | 69 (8.9) | 444 (58.6) | 240 (85.5) | 36 (89.5) | 17 (91.4) | 29 (94.6) | 29 (97.9) | 17 (99.8) | 2 (100) | | 0.5 |
| POS | | 1 (0.1) | | | | 4 (0.6) | 75 (8.9) | 334 (46.3) | 365 (87.1) | 68 (94.7) | 39 (99.1) | 3 (99.4) | 4 (99.9) | 1 (100) | 1 |
| ISA | | | 8 (1) | 45 (6) | 100 (17.2) | 300 (50.8) | 290 (83.2) | 56 (89.5) | 21 (91.8) | 24 (94.5) | 34 (98.3) | 13 (99.8) | 2 (100) | | 0.5 |
| FLC | | | | | | | 1 (0.1) | 2 (0.3) | 5 (0.9) | 31 (4.4) | 336 (41.9) | 358 (82) | 62 (88.9) | 99 (100) | 16 |
| AFG | 4 (0.4) | 3 (0.8) | 15 (2.5) | 66 (9.8) | 152 (26.8) | 370 (68.2) | 245 (95.6) | 13 (97.1) | 3 (97.4) | 12 (98.8) | 7 (99.6) | 4 (100) | | | 0.12 |
| AMB | | | | | | | | 12 (1.3) | 204 (24.2) | 676 (99.8) | 2 (100) | | | | 1 |
| **Candida krusei (120)** | | | | | | | | | | | | | | | |
| VRC | | | | | | 2 (1.7) | 29 (25.8) | 77 (90) | 11 (99.2) | 1 (100) | | | | | 0.25 |
| POS | | | | | 1 (0.8) | 2 (2.5) | 21 (20) | 54 (65) | 42 (100) | | | | | | 0.5 |
| ISA | | | | | 1 (0.8) | 8 (7.5) | 36 (37.5) | 51 (80) | 24 (100) | | | | | | 0.5 |
| AFG | | | 1 (0.8) | 4 (4.2) | 48 (44.2) | 51 (86.7) | 16 (100) | | | | | | | | 0.12 |
| AMB | | | | | | | | | 1 (0.8) | 102 (85.8) | 17 (100) | | | | 2 |
| **Candida lusitaniae (82)** | | | | | | | | | | | | | | | |
| VRC | | 8 (9.8) | 60 (82.9) | 11 (96.3) | 2 (98.8) | 0 (98.8) | 0 (98.8) | 0 (98.8) | 1 (100) | | | | | | 0.015 |
| POS | | | 1 (1.2) | 0 (1.2) | 17 (22) | 53 (86.6) | 10 (98.8) | 1 (100) | | | | | | | 0.12 |
| ISA | 7 (8.5) | 7 (17.1) | 34 (58.5) | 22 (85.4) | 11 (98.8) | | 0 (98.8) | 1 (100) | | | | | | | 0.03 |
| FLC | | | | | | | 9 (11) | 39 (58.5) | 25 (89) | 5 (95.1) | 2 (97.6) | 2 (100) | | | 1 |
| AFG | | | | | | 1 (1.2) | 9 (12.1) | 38 (58.5) | 32 (97.6) | 2 (100) | | | | | 0.5 |
| AMB | | | | | | | | 22 (26.8) | 51 (89) | 9 (100) | | | | | 1 |
| **Candida dubliniensis (107)** | | | | | | | | | | | | | | | |
| VRC | 6 (5.6) | 34 (37.4) | 53 (86.9) | 12 (98.1) | 1 (99.1) | 0 (99.1) | 1 (100) | | | | | | | | 0.015 |
| POS | | | | 8 (7.5) | 60 (63.6) | 36 (97.2) | 2 (99.1) | 1 (100) | | | | | | | 0.06 |
| ISA | 56 (52.3) | 14 (65.4) | 34 (97.2) | 1 (98.1) | 1 (99.1) | 1 (100) | | | | | | | | | 0.008 |
| FLC | | | | | 1 (0.9) | 23 (22.4) | 58 (76.6) | 22 (97.2) | 1 (98.1) | 0 (98.1) | 1 (99.1) | 0 (99.1) | 0 (99.1) | 1 (100) | 0.25 |
| ITC | | | 1 (0.9) | | 20 (19.6) | 52 (68.2) | 26 (92.5) | 7 (99.1) | 1 (100) | | | | | | 0.12 |
| AFG | | 1 (0.9) | 3 (3.7) | 11 (14) | 42 (53.3) | 28 (79.4) | 22 (100) | | | | | | | | 0.12 |
| AMB | | | | | | 2 (1.9) | 8 (9.3) | 60 (65.4) | 34 (97.2) | 3 (100) | | | | | 0.5 |
| **Candida guilliermondii (28)** | | | | | | | | | | | | | | | |
| VRC | | | | 1 (3.6) | 4 (17.9) | 14 (67.9) | 4 (82.1) | 0 (82.1) | 0 (82.1) | 1 (85.7) | 0 (85.7) | 2 (92.9) | 2 (100) | | 4 |
| POS | | | | | | | 8 (28.6) | 13 (75) | 3 (85.7) | 3 (96.4) | 1 (100) | | | | 1 |
| ISA | | | | | 2 (7.1) | 3 (17.9) | 10 (53.6) | 7 (78.6) | 1 (82.1) | 0 (82.1) | 2 (89.3) | 2 (96.4) | 0 (96.4) | 1 (100) | 4 |
| FLC | | | | | | | | | | | 12 (42.9) | 10 (78.6) | 0 (78.6) | 6 (100) | 16 |
| AFG | | | | | | | | | 2 (7.1) | 5 (25) | 19 (92.9) | 2 (100) | | | 2 |
| AMB | | | | | | | | 5 (17.9) | 20 (89.3) | 3 (100) | | | | | 1 |
| **Candida auris (7)** | | | | | | | | | | | | | | | |
| VRC | | | | | | | 3 (42.9) | 2 (71.4) | 0 (71.4) | 2 (100) | | | | | 1 |
| POS | | | | | | 4 (57.1) | 1 (71.4) | 1 (85.7) | 1 (100) | | | | | | 0.5 |
| ISA | | | | | | 1 (14.3) | 0 (14.3) | | 4 (71.4) | 2 (100) | | | | | 1 |
| FLC | | | | | | | 2 (28.6) | 5 (100) | | | | | | | 0.25 |
| AFG | | | | | | 1 (14.3) | 0 (14.3) | 6 (100) | | | | | | | 0.25 |
| AMB | | | 5 (71.4) | | | | | | | 5 (71.4) | 2 (100) | | | | 2 |

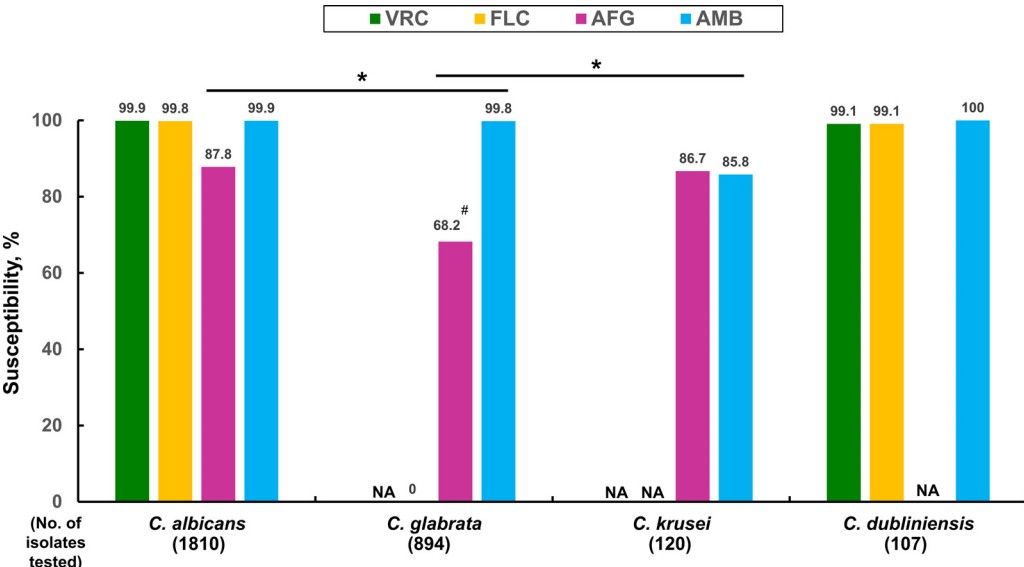

**FIG 3** *In vitro* susceptibility rates of different antifungals against isolates of important *Candida* species cultured from sterile body sites from 2017 to 2020 in the ATLAS antifungal database using the EUCAST 2020 guidelines (the numeral in parenthesis indicates the number of isolates). N/A, not applicable. *, *P* value of <0.001 when compared using Chi-square test. #, If CLSI 2020 guidelines were applied, 95.6% of *Candida glabrata* isolates tested were susceptible to anidulafungin (12).

survey (90.5%) (18). In contrast, a high *in vitro* NS rate of AMB against clinical *A. fumigatus* isolates (80.2% of which exhibited MICs of ≥4 μg/mL) was found in Hamilton, Canada in 2015 (19). The present antifungal study also revealed significantly lower *in vitro* susceptibility rates of AMB against 660 pulmonary *A. fumigatus* isolates (49.5%; Table 1) and 22 VRC-resistant *A. fumigatus* isolates (45.5%; Table 1) compared to a 2019 Spanish study (98.5% against 847 clinical *A. fumigatus sensu lato* isolates; *P* < 0.001) (20) and a 2010 to 2016 Portugal study (84.7% against 190 clinical *A. fumigatus* isolates; *P* < 0.001) (21). There is a significant diversity in the susceptibilities of AMB to *A. fumigatus* isolates collected from different countries. Consistent with the review by Posch et al. (22), *A. terreus* species complex exhibited relatively high MICs for AMB (Table 1) and high susceptibility rates to triazoles in this antifungal surveillance.

Consistent with another investigation (*r* = 0.885) (23), the MIC$_{90}$ values of ISA against IPA isolates enrolled in this study were also similar to those of VRC. A high NS rate of ISA against *A. fumigatus* isolates (22.6%) primarily collected from the thorax was reported in a 2018 to 2020 Danish survey (24). However, the ISA MIC$_{50}$ values of 0.12 to 1 μg/mL against various pulmonary *Aspergillus* species and a 7.3% NS rate of ISA against *A. fumigatus* in the present survey were similar to those of a 2012 to 2014 Denmark study (<0.125 to 2 μg/mL and 5.7%, respectively) (25). Apart from the use of agricultural triazole fungicides (8), a significantly elevated frequency of resistance to antimold triazoles in *A. fumigatus* was observed in high-temperature environments, including composts (8), greenhouses (26), and hot-climate countries such as Vietnam (27) and Pakistan (28). Further research is required to assess whether these factors contribute to the high NS rates of ISA in intrapulmonary *A. fumigatus* in Italy and the United Kingdom.

Liposomal AMB complexes have been used for the treatment of IFD for more than 20 years. The frequency of resistance to AMB in fungi was considered extremely rare because it directly binds to ergosterol, which is not genetically encoded and exerts fungicidal effects that are not physiologically favorable for developing resistance (8, 29). Due to lacking information about factors impacting AMB susceptibility, we have no idea about the cause(s) resulting in significant variations in the annual NS rates to AMB against worldwide intrapulmonary *A. fumigatus* isolates. However, Blum et al. addressed that an increased catalase activity after AMB exposure has been found in

**TABLE 4** Susceptible breakpoints of different antifungals against various *Aspergillus* and *Candida* species in the guidelines of the Clinical and Laboratory Standards Institute (CLSI) 2020 and the European Committee on Antimicrobial Susceptibility Testing (EUCAST) 2020[a]

| Species and guideline | Susceptible breakpoints ($\mu$g/mL) of different antifungal agents | | | | | | |
|---|---|---|---|---|---|---|---|
| | AMB | AFG | FLC | VRC | ISA | POS | ITC |
| *Aspergillus fumigatus* | | | | | | | |
| CLSI 2020 | N/A | N/A | N/A | 0.5 | N/A | N/A | N/A |
| EUCAST 2020 | 1 | N/A | N/A | 1 | 1 | 0.125 | 1 |
| | | | | | | | |
| *Aspergillus nidulans* spp. complex | | | | | | | |
| CLSI 2020 | N/A | N/A | N/A | N/A | N/A | N/A | N/A |
| EUCAST 2020 | N/A | N/A | N/A | 1 | 0.25 | N/A | 1 |
| | | | | | | | |
| *Aspergillus terreus* spp. complex | | | | | | | |
| CLSI 2020 | N/A | N/A | N/A | N/A | N/A | N/A | N/A |
| EUCAST 2020 | N/A | N/A | N/A | N/A | 1 | 0.125 | 1 |
| | | | | | | | |
| *Aspergillus niger* spp. complex | | | | | | | |
| CLSI 2020 | N/A | N/A | N/A | N/A | N/A | N/A | N/A |
| EUCAST 2020 | 1 | N/A | N/A | N/A | N/A | N/A | N/A |
| | | | | | | | |
| *Aspergillus flavus* spp. complex | | | | | | | |
| CLSI 2020 | N/A | N/A | N/A | N/A | N/A | N/A | N/A |
| EUCAST 2020 | N/A | N/A | N/A | N/A | 1 | N/A | 1 |
| | | | | | | | |
| *Candida albicans* | | | | | | | |
| CLSI 2020 | N/A | 0.25 | 2; (SDD[b]) 4 | 0.12 | N/A | N/A | N/A |
| EUCAST 2020 | 1 | 0.03 | 2 | 0.06 | 0.06 | 0.06 | 0.06 |
| | | | | | | | |
| *Candida glabrata* | | | | | | | |
| CLSI 2020 | N/A | 0.12 | N/A | N/A | N/A | N/A | N/A |
| EUCAST 2020 | 1 | 0.06 | 0.001 | N/A | N/A | N/A | N/A |
| | | | | | | | |
| *Candida krusei* | | | | | | | |
| CLSI 2020 | N/A | 0.25 | N/A | 0.5 | N/A | N/A | N/A |
| EUCAST 2020 | 1 | 0.06 | N/A | N/A | N/A | N/A | N/A |
| | | | | | | | |
| *Candida dubliniensis* | | | | | | | |
| CLSI 2020 | N/A | N/A | N/A | N/A | N/A | N/A | N/A |
| EUCAST 2020 | N/A | N/A | 2 | 0.06 | 0.06 | 0.06 | 0.06 |
| | | | | | | | |
| *Candida guilliermondii* | | | | | | | |
| CLSI 2020 | N/A | 2 | N/A | N/A | N/A | N/A | N/A |
| EUCAST 2020 | N/A | N/A | N/A | N/A | N/A | N/A | N/A |

[a]The susceptible breakpoints of antifungals against *Candida lusitaniae* and *Candida auris* are lacking in the both guidelines in 2020. N/A, not applicable.
[b]SDD, susceptible dose dependent.

association with AMB resistance in *Aspergillus* species (19, 30). Additionally, through the analyses of single nucleotide polymorphisms, Fan et al. found that mutations on six kinds of genes (*ERG3*, *TcsB*, *MpkB*, *MpkC*, *CatA*, *Fos1*) in *A. fumigatus* are significantly associated with decreased ergosterol biosynthesis or defective reduction-oxidation homeostasis, conferring them resistant to AMB (31). The maximal concentration ($C_{max}$)/MIC ratio of 2.5 for AMB, an important PD target of AMB against *A. fumigatus*, is associated with near-maximal antifungal efficacy in a murine IPA model (32). The liposomal AMB complex, which is well-localized at lung infection sites, was shown to be considerably superior *in vivo* to deoxycholate AMB with regard to improving survival in a murine IPA model (33). However, according to PK data on the concentration of liposomal AMB complex in blood (liberated form, 1.08 $\pm$ 0.31 $\mu$g/mL) and its concentration at epithelial lining fluid (ELF) (1.60 $\pm$ 0.58 $\mu$g/mL) following administration with a dosage of 4 to 5 mg/kg/day for 5 to 7 days (Table 5) (34), a prescription of liposomal AMB complex with a dosage of >4 mg/kg/day for the treatment of IPA isolates with MICs of >0.5 $\mu$g/mL is estimated to be likely ineffective. This result corresponded to the findings of a low susceptibility rate against all *A. fumigatus* isolates (<50%) as assessed

**TABLE 5** Pharmacokinetic profiles and pharmacodynamic indices of antifungal agents (antimold triazoles, anidulafungin, and liposomal amphotericin B complex) against the studied molds[a]

| Antibiotic and dosage (PD indices against molds) | PK profile | | | | | |
| --- | --- | --- | --- | --- | --- | --- |
| | $C_{max}$ ($\mu$g/mL), total form drug | Half-life (h) | Protein binding (%) | $fAUC_{0-24}$ (mg · h/L) | $C_{ELF}$ ($\mu$g/mL) | ELF/plasma ratio |
| Voriconazole (%$T$>MIC) | 7~8 (6 mg/kg bid on day 1, then 4 mg/kg bid since day 2) | 6 | 58 | Mean, 24.8 (4 mg/kg) | Mean, 19.3 | Mean, 11 (±8), calculated in terms of the concentrations |
| Posaconazole 400 mg bid (%$T$>MIC) | Mean, 3.2 (multiple doses of 400 mg bid) | 24~27 | 98 | 73 | | 0.86~1.02 in terms of AUC ratio; 0.20~0.31 in terms of total concentrations |
| Isavuconazole 372 mg every 8 h × 6 doses, and 372 mg once daily since day 3 (%$T$>MIC) | 245~345 (mean, 295~300) | 85~117 | 99 | 97 | | 0.36~0.73 in terms of AUC ratio |
| Anidulafungin 100 mg once daily following 200 mg loading (%$T$>MIC, $C_{max}$) | 7.0~7.2 | 40~50 | 99 | 98 | 0.9~1.1 | 0.15~0.37 |
| Liposomal amphotericin B complex ($C_{max}$/MIC) | 5.17 (±1.89, for total form); 1.08 (±0.31, for liberated form) (multiple doses of 4.55 ± 0.23 mg/kg/day) | 400 | 98 | 168 (3 mg/kg/day) | 1.60 (± 0.58) (4.55 ± 0.23 mg/kg/day) | 1.54 (± 0.44) for liberated form |

[a]$C_{max}$, maximal concentration in serum; bid, twice daily; $fAUC_{0-24}$, area under the concentration-time curve from 0 to 24 h; ELF, epithelial lining fluid; $C_{ELF}$, concentration in epithelial lining fluid; %$T$>MIC, the percentage of a dosage interval in which the serum level exceeds the MIC of a given organism.

using the EUCAST 2020 guidelines (Table 1). In addition, although the susceptibility MIC breakpoints of AMB against *A. flavus* and *A. terreus* species complex are currently lacking in the EUCAST 2020 guidelines (13), according to Al-Saigh et al.'s *in vitro* investigation (35), we estimate that the liposomal AMB complex is also less effective against *A. flavus* and *A. terreus* species complex, which requires significantly higher $C_{max}$s for effective killing than that against *A. fumigatus* (35).

An analysis of POS concentration monitoring in IPA treatment showed a strong clinical concentration-efficacy relationship (36). In this survey, the susceptibility rate of POS to intrapulmonary *A. fumigatus* isolates (18.9%; Table 1) was significantly different from that in a Spanish investigation (94 to 94.5%; $P < 0.001$) (20). However, the PK characteristics of POS (Table 5) include the following: the protein affinity was 98%, the half-life was 24 to 27 h (37), the mean plasma $C_{max}$ was 3.2 $\mu$g/mL following administration of multiple doses of 400 mg twice daily (38), the ratio of penetration from plasma to ELF ranged from 0.20 to 0.31 in terms of total concentrations (39), and the mean value of the area under the concentration-time curve (AUC) from 0 to 12 h ($AUC_{0-12}$) in ELF ranged from 11.2 to 18.3 mg · h/L following oral administration of multiple doses of 400 mg twice daily in humans (40, 41). Furthermore, a supplementary PD target of the AUC/MIC ratio achieving 1-log killing of *A. fumigatus* was 2.07 $\pm$ 1.02 (6). Through the POS's PK and PD data, we estimate that POS administered orally at multiple doses of 400 mg twice daily is likely an effective regimen for the treatment of intrapulmonary *Aspergillus* isolates with MICs of <0.5 $\mu$g/mL. Markedly contrasting with a low *in vitro* susceptibility against *A. fumigatus* (18.9%; Table 1), as assessed by the EUCAST 2020 criteria (13) from the aforementioned PK/PD data and MIC distributions, POS is a reliable salvage therapy against important pulmonary *Aspergillus* species as recommended by the ESCMID-ECMM-ERS guidelines. Similar to POS and consistent with the favorable treatment outcomes of IPA patients receiving therapy of ISA (MICs of $\geq$2 $\mu$g/mL for the implicated intrapulmonary *Aspergillus* isolates, day 42 mortality rate, 21.4% [3/14]) in the SECURE study (42), an excellent PK profile as well as PD index and good tolerability were observed for ISA (43–45) against intrapulmonary isolates of *Aspergillus* species, including ISA-resistant *Aspergillus* isolates (Table 5).

Noncomparative studies have shown that therapy with capsofungin achieved a 30 to 90% success rate in the clinical treatment of IPA (46). According to a report published by Li et al. (37), the percentage of dosage interval in which the serum level exceeds the MIC (i.e., %$T_{>MIC}$) and the $C_{max}$ at the infection sites (including ELF and alveolar macrophages [AM]) are critical determinants of the fungistatic activity of AFG. In this survey, the $MIC_{90}$ values of AFG against most intrapulmonary *Aspergillus* isolates ranged from 0.016 to 0.03 $\mu$g/mL (Table 2). Following intravenous administration of 200 mg loading and 100 mg once daily for more than 10 days, the AFG's PK characteristics include the following: the protein affinity was 99% (47), the half-life was 40 to 50 h, total drug $C_{max}$ was 7.0 to 7.2 $\mu$g/mL (48), and the AFG concentrations at ELF and AM were 0.9 to 1.1 and $\geq$37 $\mu$g/mL, respectively (49). According to the AFG's PD parameters and MIC distributions against intrapulmonary isolates of *Aspergillus* species enrolled in this ATLAS study, we estimate that the standard-dose AFG regimen is likely to be potent against important IPA isolates. Nevertheless, the clinical efficacy of AFG monotherapy or in combination with other antifungals (AMB or VRC) against IPA in patients with hematological cancer is debatable (50).

This *in vitro* antifungal study has some limitations. First, some single-point alterations (G138C, Y431C, G434C, and G448S, etc.) (23) and other mutations involving $TR_{34}$/L98H, $TR_{34}$/L98H/S297T/F495I, and $TR_{46}$/Y121F/T298A, among others, in the *CYP51A* gene (encoding 14-$\alpha$-demethylase, the main target of triazoles) sequence (8, 24, 25) were shown to be responsible for VRC resistance in *A. fumigatus* isolates; however, these VRC-resistant *A. fumigatus* isolates are not available for the resistance gene analyses. Second, the number of intrapulmonary *A. fumigatus* isolates collected from some ATLAS participant countries might be too low to perform susceptibility analysis. Third, the clonality of the VRC-resistant *A. fumigatus* isolates in this survey was not analyzed.

In conclusion, this *in vitro* 2017 to 2020 ATLAS investigation regarding the susceptibility profile of clinically important fungi showed high NS rates for AMB and POS against isolates of important IPA species. In contrast, ITC, ISA, and VRC were highly active *in vitro* against most IPA isolates worldwide, except in Italy and the United Kingdom. From the PK/PD profile of AFG and the relevant MIC distributions of AFG against IPA, we consider that this antifungal agent may have good potency against important IPA species, although its clinical anti-IPA efficacy needs further deliberative evaluation. These data highlight the need for continued surveillance of the *in vitro* activities of the available antifungals.

## MATERIALS AND METHODS

**ATLAS protocol.** In 2006, Pfizer Pharmaceutical (New York City, NY, USA) conducted the ATLAS project to investigate the global *in vitro* susceptibility data of important microorganisms implicated in clinical infections. The National Taiwan University Hospital (NTUH) (Taipei, Taiwan), a 2,500-bed tertiary care center in northern Taiwan, has participated in the ATLAS program since 2012. The ATLAS program was approved by the institutional review board of each participating center, including NTUH (NTUH 201211047RSC).

**Enrolled fungal isolates.** Most *Candida parapsilosis* and *Candida tropicalis* isolates show excellent *in vitro* susceptibility to FLC (16). Other than *C. parapsilosis* and *C. tropicalis*, in the ATLAS antifungal survey, we enrolled isolates of some clinically important *Candida* species (including *C. albicans*, *C. auris*, *C. dubliniensis*, *C. glabrata*, *C. guilliermondii*, *C. krusei*, and *C. lusitaniae* [13, 16]) that were collected from sterile body sites, including bone tissue, other biopsied tissue, blood, and fluid aspirated from the intra-articular, cerebrospinal, peritoneal, or pleural space. In addition, all enrolled *Aspergillus* isolates were collected from secretions of the lower respiratory tract (expectorated sputum, tracheal aspirate, and bronchoalveolar lavage or wash fluid) or the biopsied tissue of the lung parenchyma. Because the MIC data of ISA against fungi (including important *Candida* and *Aspergillus* species) collected from the ATLAS database have been available since 2017, the 2017 to 2020 antifungal MIC results in the ATLAS database were extracted for analysis in this study.

**Species identification.** Sabouraud dextrose agar (SDA) was used for the culture of *Aspergillus* species. In addition, SDA, Sabouraud brain heart infusion agar, or both were used for culture of *Candida* species. Species were first identified using macro- and micromorphology at each participating center. The species were further confirmed using thermotolerance (incubation at 37°C and 43°C) and matrix-assisted laser desorption ionization–time of flight mass spectrometry (Bruker, Bremen, Germany) for *Candida* species and using thermotolerance (incubation at 50°C) for *A. fumigatus* isolates, respectively, at the International Health Management Associates (Schaumburg, IL, USA) prior to susceptibility testing. In this study, the term "species complex" for some *Aspergillus* species other than *A. fumigatus* in the absence of detailed molecular characterization was used as stated elsewhere (25). According to the ATLAS protocol, only the first isolate from a single patient was included in the ATLAS program.

**Criteria for susceptibility interpretation.** Broth microdilution susceptibility testing was used for *Aspergillus* and *Candida* species in accordance with the CLSI M38 and CLSI M51 standards, respectively (51, 52). As antimold triazole-resistant *A. fumigatus* isolates have been detected in many regions worldwide (7), we also explored the susceptibility of antifungals against VRC-resistant *A. fumigatus* isolates in this survey. The concentration range tested for antifungals was 0.002 to 16 $\mu$g/mL.

**Statistical analysis.** Categorical variables are expressed as percentages of the total number of isolates. The differences in susceptibility rates between antifungals against *Aspergillus* and *Candida* species were analyzed using the chi-square test as appropriate. When the susceptibility MIC breakpoints against specific *Aspergillus* species were lacking, the MIC distributions of a given antifungal against different isolates of *Aspergillus* species were compared using the Mann-Whitney U test as appropriate. Pearson's correlation analysis and calculation of variance (standard deviation) were used to analyze the trend and degree of deviation, respectively, regarding the annual susceptibility rates of the three antifungals against intrapulmonary *A. fumigatus* isolates collected between 2017 and 2020. All statistical calculations were two-tailed, and a *P* value of $<0.05$ was considered statistically significant. All statistical analyses were performed using SPSS version 17.0 (Chicago, Armonk, NY, USA).

**Ethical approval.** The institutional review board of each participating center, including the National Taiwan University Hospital (Taipei, Taiwan) [NTUH 201211047RSC], approved the ATLAS study.

## ACKNOWLEDGMENTS

This study was supported by Pfizer Pharmaceutical (New York, NY).

We declare no competing interests.

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
