## [Reviewer comments · Microbiology Spectrum]

Microbiology Spectrum

In Vitro Susceptibilities of Worldwide Isolates of Intrapulmonary *Aspergillus* Species and Important *Candida* species in Sterile Body Sites against Important Antifungals: Data from the Antimicrobial Testing Leadership and Surveillance Program, 2017–2020

Shio-Shin Jean, Hung-Jen Yang, Po-Chuen Hsieh, Yu-Tsung Huang, Wen-Chien Ko, and Po-Ren Hsueh

Corresponding Author(s): Po-Ren Hsueh, China Medical University Hospital

Review Timeline:

Submission Date:	August 11, 2022
Editorial Decision:	September 8, 2022
Revision Received:	September 18, 2022
Accepted:	September 27, 2022

Editor: Joshua Obar

Reviewer(s): The reviewers have opted to remain anonymous.

Transaction Report:

DOI: <https://doi.org/10.1128/spectrum.02965-22>

September 8, 2022

Prof. Po-Ren Hsueh
China Medical University Hospital
Laboratory Medicine and Internal Medicine
No 7 Chung-Shan South Road
Taichung, Taiwan 404394
Taiwan

Re: Spectrum02965-22 (In Vitro Susceptibilities of Worldwide Isolates of Intrapulmonary Aspergillus Species and Important Candida species in Sterile Body Sites against Important Antifungals: Data from the Antimicrobial Testing Leadership and Surveillance Program, 2017-2020)

Dear Prof. Po-Ren Hsueh:

Thank you for submitting your manuscript to Microbiology Spectrum. As you will see your paper is very close to acceptance. Please modify the manuscript along the lines I have recommended. As these revisions are quite minor, I expect that you should be able to turn in the revised paper in less than 30 days, if not sooner. If your manuscript was reviewed, you will find the reviewers' comments below.

Editors Note: There is considerable enthusiasm for your manuscript from everyone involved as this is an important and timely topic of investigation. Everyone agrees the datasets are robust and important to the field. There are just some suggestions about how to improve the readability of the manuscript to ensure the important datasets and messages get presented as well as possible to the readers.

When submitting the revised version of your paper, please provide (1) point-by-point responses to the issues raised by the reviewers as file type "Response to Reviewers," not in your cover letter, and (2) a PDF file that indicates the changes from the original submission (by highlighting or underlining the changes) as file type "Marked Up Manuscript - For Review Only". Please use this link to submit your revised manuscript. Detailed instructions on submitting your revised paper are below.

Link Not Available

Sincerely,

Joshua Obar

Reviewer comments:

Reviewer #1 (Comments for the Author):

Thanks extended to the authors for carrying out this analysis. It is important to know the prevalences of resistance rates and antifungal susceptibility across the world. I have a few suggestions to improve the manuscript, which will improve the readability.

I find the introduction a bit disjointed - it starts with "septicemia can cause life-threatening complications", and then goes on to talk about invasive candidiasis and candidemia, and later invasive pulmonary aspergillosis, without linking back to septicemia at all.

I do not agree that 'periodic monitoring of fungal species-specific susceptibility is beneficial for understanding the evolution of antifungal resistance'; whilst monitoring of susceptibility is needed, it tells you nothing about how resistance has evolved, or the mechanisms behind it. It does, however, tell you about the changes in prevalence temporally and spatially. This phrase of 'evolutionary changes' is mentioned again on page 8 when referring to Figure 1, which shows the changes in prevalence of

susceptibility, and not the evolutionary changes responsible for this. Indeed, the legend of Figure 1 actually describes these data accurately, and this is how it should be referred to throughout the paper.

Consider using 'Czechia' instead of the Czech Republic, as it wishes to be known by the former.

Pharmacokinetic profiles of the antifungal agents are only mentioned in the discussion; there is no analysis of these in relation to the findings in the results (in the introduction it is stated that "PK profiles" will be "assessed" - but they aren't. They're just provided in the discussion).

Also, the discussion mentions that single point alterations and other mutations were not identified in VRC-resistant *A. fumigatus* isolates - these data need to be included in the results, and also how they were identified included in the methods.

Reviewer #2 (Comments for the Author):

This manuscript by Jean et al. summarized the susceptibility of *Aspergillus* and *Candida* isolates collected in clinics worldwide between 2017-2020. Studies that summarize susceptibility trends are important for monitoring the continued emergence of drug resistance among clinical isolates. The result presented here align well with previous studies from specific geographical regions however this study highlights interesting trends that have not previously reported. The following comments should be considered:

1. This study would be strengthened by defining the cutoffs used for determining susceptibility in tables 1-3 and subsequent figures. This will not only help readers who aren't familiar with specific cutoffs for each organism/drug but will also facilitate comparison with future studies where susceptibility cutoffs may change as more clinical outcome data is compiled for these treatments.

2. Lines 132-125 "During the four year study period..." The authors mention a wider fluctuation of AMB susceptibility than ISA or POSA and in figure 1 show much lower susceptibility in 2019 and 2020 compared to 2017 and 2018. It would be interesting to comment in the text on whether these trends are driven by large changes in certain regions or specific countries or if resistance patterns changed globally between these years.

3. A few grammatical changes

Line 74 "immunity" changed to "immune"

Line 76 "prescriptions" changed to "prescription"

Line 78 "virus" deleted

Line 211 "one-doubling dilutions" changed to "two-fold"

Line 363 "antifungal antibiotics" changed to "antifungals"

Reviewer #3 (Comments for the Author):

Jean and collaborators evaluated the in vitro antifungal susceptibilities against clinically important isolates of *Aspergillus* and *Candida* species in an antifungal resistance Surveillance program. The data obtained in this work will contribute to the epidemiology of resistance to antifungal agents.

Preparing Revision Guidelines

Please return the manuscript within 60 days; if you cannot complete the modification within this time period, please contact me. If you do not wish to modify the manuscript and prefer to submit it to another journal, please notify me of your decision immediately so that the manuscript may be formally withdrawn from consideration by Microbiology Spectrum.

Responses to the Reviewers' Comments

Reviewer #1 (Comments for the Author):

Thanks extended to the authors for carrying out this analysis. It is important to know the prevalences of resistance rates and antifungal susceptibility across the world. I have a few suggestions to improve the manuscript, which will improve the readability.

I find the introduction a bit disjointed - it starts with "septicemia can cause life-threatening complications", and then goes on to talk about invasive candidiasis and candidemia, and later invasive pulmonary aspergillosis, without linking back to septicemia at all.

Reply: We have revised this sentence accordingly.

I do not agree that 'periodic monitoring of fungal species-specific susceptibility is beneficial for understanding the evolution of antifungal resistance'; whilst monitoring of susceptibility is needed, it tells you nothing about how resistance has evolved, or the mechanisms behind it. It does, however, tell you about the changes in prevalence temporally and spatially. This phrase of 'evolutionary changes' is mentioned again on page 8 when referring to **Figure 1**, which shows the changes in prevalence of susceptibility, and not the evolutionary changes responsible for this. Indeed, the legend of **Figure 1** actually describes these data accurately, and this is how it should be referred to throughout the paper.

Reply: We thank the reviewer's valuable suggestion. We have revised them accordingly.

Consider using '**Czechia**' instead of the Czech Republic, as it wishes to be known by the former.

Reply: We have revised them in the text and edited **Figure 2** accordingly.

Pharmacokinetic profiles of the antifungal agents are only mentioned in the discussion; there is no analysis of these in relation to the findings in the results (in the introduction it is stated that "PK profiles" will be "assessed" - but they aren't. They're just provided in the discussion).

Reply: In fact, using the mathematical calculations according to the respective PK profile and required PD parameter(s), we have evaluated the suitability of

currently recommended antifungal doses against the tested intrapulmonary isolates of specific *Aspergillus* species by means of the MIC distributions and susceptibility data in the ATLAS that were presented in the original manuscript. The relevant statements are as follows --- **Liposomal AMB complex** vs. *A. fumigatus*: included in line 296-298; **liposomal AMB complex** vs. *A. flavus/A. terreus*: included in line 303-306; **posaconazole** vs. all intrapulmonary *Aspergillus* species: included in line 320-322; **anidulafungin** vs. all intrapulmonary *Aspergillus* isolates: included in line 345-348; and **isavuconazole** vs. intrapulmonary *A. fumigatus* isolates: included in line 327 and 330-333 of the revised manuscript.

Also, the discussion mentions that single point alterations and other mutations were not identified in VRC-resistant *A. fumigatus* isolates - these data need to be included in the results, and also how they were identified included in the methods.

Reply: We apologized for being unable to provide the results regarding the genetic mutations in voriconazole (VRC)-resistant *A. fumigatus* isolates because these VRC-resistant isolates are not available for genetic analysis. Consequently, we have revised the relevant statement in the **Discussion** section of the revised manuscript. Additionally, we will not add the methods employed for the analysis of resistance genes in intrapulmonary VRC-resistant *A. fumigatus* isolates in the revised manuscript as well.

Reviewer #2 (Comments for the Author):

This manuscript by Jean et al. summarized the susceptibility of *Aspergillus* and *Candida* isolates collected in clinics worldwide between 2017-2020. Studies that summarize susceptibility trends are important for monitoring the continued emergence of drug resistance among clinical isolates. The result presented here align well with previous studies from specific geographical regions; however, this study highlights interesting trends that have not previously reported. The following comments should be considered:

1. This study would be strengthened by defining the cutoffs used for determining susceptibility in tables 1-3 and subsequent figures. This will not only help readers who aren't familiar with specific cutoffs for each organism/drug but will also facilitate comparison with future studies where susceptibility cutoffs may change as more clinical outcome data is compiled for these treatments.

Reply: For strengthening the understanding of the *in vitro* susceptibility cutoff points of various antifungals, we decide to add a new Table (**Table 4**) which lists the susceptible breakpoints of different antifungals against various *Aspergillus* species and *Candida* species according to the CLSI 2020 and EUCAST 2020 guidelines in the revised manuscript.

2. Lines 132-125 "During the four year study period..." The authors mention a wider fluctuation of AMB susceptibility than ISA or POSA and in **Figure 1** show much lower susceptibility in 2019 and 2020 compared to 2017 and 2018. It would be interesting to comment in the text on whether these trends are driven by large changes in certain regions or specific countries or if resistance patterns changed globally between these years.

Reply: We indeed have no idea about this significant variation in annual AMB susceptibility rates against intrapulmonary *A. fumigatus* isolates. Hence, we have added this comment and relevant information regarding the decreased AMB susceptible rates in *A. fumigatus* which was observed by other investigators to the text of the revised manuscript.

3. A few grammatical changes

Line 74 "immunity" changed to "immune"

Line 76 "prescriptions" changed to "prescription"

Line 78 "virus" deleted

Line 211 "one-doubling dilutions" changed to "two-fold"

Line 363 "antifungal antibiotics" changed to "antifungals"

Reply: We have revised them accordingly.

Reviewer #3 (Comments for the Author):

Jean and collaborators evaluated the in vitro antifungal susceptibilities against clinically important isolates of *Aspergillus* and *Candida* species in an antifungal resistance Surveillance program. The data obtained in this work will contribute to the epidemiology of resistance to antifungal agents.

Reply: We greatly appreciated the reviewer's comments.

September 27, 2022

Prof. Po-Ren Hsueh
China Medical University Hospital
Laboratory Medicine and Internal Medicine
No 7 Chung-Shan South Road
Taichung, Taiwan 404394
Taiwan

Re: Spectrum02965-22R1 (In Vitro Susceptibilities of Worldwide Isolates of Intrapulmonary Aspergillus Species and Important Candida species in Sterile Body Sites against Important Antifungals: Data from the Antimicrobial Testing Leadership and Surveillance Program, 2017-2020)

Dear Prof. Po-Ren Hsueh:

Your manuscript has been accepted, and I am forwarding it to the ASM Journals Department for publication. You will be notified when your proofs are ready to be viewed.

Sincerely,

Joshua Obar
Editor, Microbiology Spectrum
